# Potential Uses of Artisanal Gold Mine Tailings, with an Emphasis on the Role of Centrifugal Separation Technique

Jeanne Pauline Munganyinka [1,2,*], Jean Baptiste Habinshuti [1,2], Jean Claude Ndayishimiye [3,4], Levie Mweene [5], Grace Ofori-Sarpong [1,6], Brajendra Mishra [2,*], Adelana R. Adetunji [1,7] and Himanshu Tanvar [2]

1. Department of Materials Science and Engineering, African University of Science and Technology, Abuja 900100, Nigeria; hbaptiste@aust.edu.ng (J.B.H.); gofori-sarpong@umat.edu.gh (G.O.-S.); aderade2004@yahoo.com (A.R.A.)
2. Mechanical Engineering, Worcester Polytechnic Institute, Worcester, MA 01609, USA; htanvar@wpi.edu
3. Department of Biology, Shenzhen MSU-BIT University, Shenzhen 518172, China; ndayiclaude2006@yahoo.fr
4. The Center for Earth and Natural Resource Sciences, Kigali P.O. Box 4285, Rwanda
5. Department of Materials Engineering, Indian Institute of Science, Bangalore 560012, India; leviemweene@alum.iisc.ac.in
6. Department of Minerals Engineering, University of Mines and Technology, Tarkwa P.O. Box 237, Ghana
7. Department Materials Science and Engineering, Obafemi Awolowo University, Ile-Ife 220282, Nigeria
* Correspondence: jmunganyinka@aust.edu.ng (J.P.M.); bmishra@wpi.edu (B.M.); Tel.: +1-570-664-4338 (J.P.M)

**Abstract:** Few investigations have focused on the potential uses of artisanal gold (Au) mine tailings, despite the fact that artisanal gold mining activity contributes to environmental issues such as greenhouse gas. Mineralogical characterizations of artisanal gold mine tailings in Miyove gold mine (Baradega and Masogwe) in Rwanda were investigated for potential utilization as a source of valuable gold, using the centrifugal separation technique. Results of X-ray diffraction analysis, energy dispersive X-ray spectroscopy, inductively coupled plasma mass spectrometry, inductively coupled plasma–optical emission spectroscopy, and X-ray fluorescence showed that artisanal gold mine tailings samples have significant amounts of gold to justify economical gold extraction opportunity. The gold grades in the ores and artisanal gold mine tailings were in the ranges of 37–152 and 2–7 g t$^{-1}$, respectively. Quartz was a major phase, with minor impurities in two different types of gold ores and their respective tailings. The beneficiation carried out using centrifugal separation, regarded as an extension of gravity separation, showed gold grades in the range of 535–1515 g t$^{-1}$ for gold ores and 36–302 g t$^{-1}$ for artisanal gold mine tailings. The gold recoveries for ores and artisanal gold mine tailings were in the range of 21.8–47.3% and 46.9–63.8%, respectively. The results showed that the centrifugal separation technique was more efficient in boosting gold recovery compared to the present panning approach employed at the site, which sometimes recover as low as 10%. The results suggest that mineralogical characterization of artisanal gold mine tailings allows for the development and design of a suitable methods for improving gold ore beneficiation and artisanal gold mine tailings reprocessing.

**Keywords:** gold mining; artisanal gold mine tailings; centrifugal separation; mineralogical characterization; artisanal and small-scale mining

## 1. Introduction

Gold has been prized for its beauty and durability since ancient times [1]. Rwanda is a major producer of tin, tantalum, and tungsten (3Ts), as well as a gold and gemstone exporter [2]. Mineral exports brought in approximately USD 67 million in 2010, accounting for around 15% of total exports and making mining the country's main source of foreign earnings [3,4]. Due to the high global gold demand, the gold mining industry has recently

emerged to have great potential, but most of this is carried out by individuals or cooperatives (artisanal small-scale miners) [5,6]. The gold mine tailings produced by small-scale miners are often dumped in designated tailing ponds with minimal treatment [5,7,8]. An increased artisanal gold mining in Rwanda has caused a rapid increase in artisanal gold mine tailings dumped yearly, and this can pose substantial risks to the health of people living around the mine sites as well as the neighboring ecosystems [9]. The mismanagement of the disposed tailings results in sliding, deforestation, and disturbance in the water table channel. In addition to decontaminating these mine tailings to prevent the release of harmful components, potential uses for these large quantities of mine tailings must be consistently evaluated for metal extraction [10,11]. Thus, re-processing of artisanal gold mine tailings is a sustainable way of minimizing the consequences of these wastes on the environment.

In nature, most gold occurs as nuggets and free grains in rocks, veins, and alluvial deposits [12]. Each gold deposit has unique mineralogical characteristics, thus requiring a specific type of processing to obtain pure gold [12–17]. The choice of beneficiation process depends on the nature of the gangue present in the ore and its association with the mineral of economic value [13,18–20]. For that reason, it is important to carry out a geometallurgical and diverse characterization analysis, which allows for a better observation of the different characteristics of the ore and its impact on the treatment of generated tailings [17,21,22]. The steps commonly employed in the processing of raw ore include crushing and grinding, screening, classification, concentration, dewatering, and disposal of tailings [23–25]. For the concentration and treatment of tailings, the steps commonly employed are classified using spirals, hydrocyclones, magnetic concentration, and flotation [14,26,27]. Chemical and mineralogical characterization allow for a better observation of the different mineralogical characteristics of gold ore and their relationship with the processing steps [28,29]. Grayson (2007) [30] identified about 70 different gold processing methods, except those based on cyanide or mercury, to recover gold lost by artisanal miners without side effects of cyanide and mercury. With the focus of avoiding the side effects caused by the mercury and cyanide, various gravity separation techniques have been tested to increase their suitability in artisanal gold mine [31]. These approaches allow the suggestion of different processing techniques, leading to changes in the processing steps already utilized. The inclusion of the information obtained by the mineral characterization converges to the optimization of mine development and, therefore, to the reduction of gold lost in the tailings, proving a better use of the mineral resources of a deposit. Thus, mineral characterization is of great importance for the future of mining since the comprehensive use of artisanal gold mine tailings is vital for the conservation of natural resources and, ensuring the sustainable development of the mining industry.

This research is therefore focused on the characterization and beneficiation of a gold ore, sourced from artisanal gold mine tailings. In light of this, a study on the mineralogical characteristics of both gold ores and resultant artisanal gold mine tailings was performed using different analytical tools. To achieve the best grade and recovery, gold ore was beneficiated, and artisanal gold mine tailings were treated using gravity concentration. This study is motivated by the search for alternatives that boost the re-treatment of tailings. The goal of this research was to perform a mineralogical characterization of artisanal gold mine tailings from a mine located in Rwanda and evaluate the technical feasibility of reprocessing this material and using it as a potential secondary source of gold.

## 2. Materials and Methods

### 2.1. Site Description and Sampling

The gold ores and artisanal gold mine tailings used in this work were collected from two zones (Baradega and Masogwe) of a mine in the region of Miyove in northern Rwanda (Figure 1). The topography of the mine area is defined by the steep hills and flat, marshy ground in the valleys. The elevation ranges from 1700 to 2200 m above sea level. Details of the local processing method (Figure S1) and regional geology of Miyove and the history

of artisanal gold mining are shown in the Supplementary information. The samples were the representative fresh quartz vein of the gold ore deposit and disposed artisanal gold mine tailings that were mixed using cone and quartering methods. Two gold ore samples ($n$ = 2, mass = 10 kg) and disposed artisanal gold mine tailings ($n$ = 2, mass = 10 kg) were packed in labeled polyethylene plastic bags and immediately returned to the laboratory for further analysis. Each ore sample collected from the mine was crushed with a hammer to generate particles with a diameter of <40 mm. To produce particles with a diameter of 0.5 mm from those with <40 mm, the RETSCH 200520035 jaw crusher BB 50 Tungsten, which allows the user to set up a desired particle size, was used. A cone and quartering sampling were utilized to obtain an evenly dispersed sample. A ball mill was used to grind evenly dispersed samples to finer particles of a diameter ≤75 μm. After grinding, the sample was mixed thoroughly with the aid of the riffle sample splitter, and a representative sample was obtained for physico–chemical analysis and mineralogical characterization. For the process of beneficiation, a sieve shaker was used to produce particles with a diameter in the range of 45 to 600 μm. The particles with a diameter above 200 μm and the collected tailings were also ground by using a ball mill for the beneficiation process.

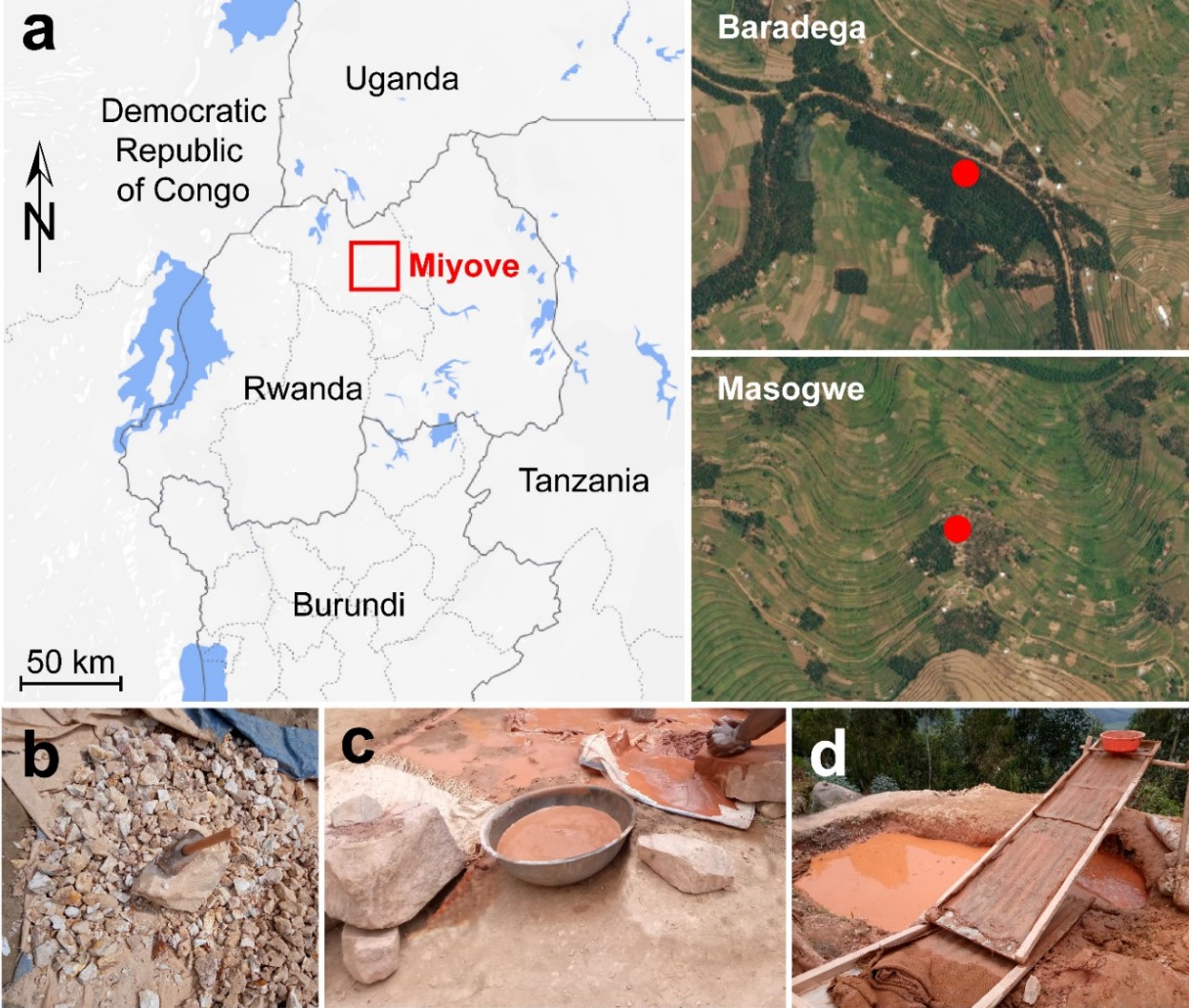

**Figure 1.** Artisanal gold mining. (**a**) The location of the Miyove gold mine and landscape of two sampling sites (red points). The photos of three artisanal gold mining methods at Miyove: (**b**) gold ore breaking with Hammer; (**c**) African stone grinding mill; (**d**) old-style table shaking system.

### 2.2. Mineralogy and Analysis of Concentrate

The particle size and shape analyses were carried out using a sieve shaker with meshes in the range of 45–600 µm and a Microtrac FlowSync (Microtrac Retsch GmbH, Haan/Duesseldorf, Germany). The mineralogical characterization of the ore samples ($n = 2$) and tailings samples ($n = 2$) was carried out using an X-ray diffractometer (Malvern PANalytical, Westborough, MA, USA) with a Cu-K$\alpha$ ($\lambda = 0.1540$ nm) radiation source operated at 45 kV and 40 mA. X'Pert HighScore Plus version 4.6a was used to identify the phases present in the samples. The chemical characterization of samples was performed using scanning electron microscopy in combination with energy-dispersive X-ray spectroscopy (JEOL JSM-700F, Hollingsworth & Vose, East Walpole, MA, USA). The solid samples were digested in an aqua regia solution (AR, 1:3 ratio of $HNO_3$ and HCl) to analyze the gold content with inductively coupled plasma mass spectrometry (ICP-MS, PerkinElmer, NEXION350x) [32]. For the quantitative elemental analysis, an inductively coupled plasma–optical emission spectroscopy (ICP-OES, Perkin Elmer Optima 8000) and X-ray fluorescence analyzer (XRF, Olympus Vanta M-Series) were also used. The samples for the ICP-OES were prepared by fusing the sample ore (0.1 g) with lithium tetraborate (1.0 g) as fluxing material in the graphite crucible at 1000 °C for 1 h. Then, 25% nitric acid was used to dissolve the melt and 2% of nitric acid was used to dilute the solution and also as a blank for ICP-OES analyses [33]. All analyses were carried out in triplicate, and the average values were reported.

### 2.3. Centrifugal Separation Technique

The procedures followed in the centrifugal separation of gold ores and tailings are depicted in Figure 2. In brief, beneficiation tests were carried out using a centrifugal concentrator blue bowl customized with kit w/pump and leg levelers in the presence of water as a fluid (Figure 2b). The riffle sample splitter was used to obtain homogeneous samples. Three beneficiation trials of 10 kg of gold ores and 10 kg of artisanal gold mine tailings of size <200 µm were processed. After cleaning the bowl and ensuring that all connected devices are in place and well controlled, a 12-volt pump was used to pump water into the bowl. The separation process uses centrifugal forces created by the vortex movement of water as it moves around the inner cone. Carefully ground samples were dropped into the bowl and the control valve was opened slowly to avoid overflow. As the water moved around, the inner cone carried the lighter materials to the surface, dropped them into a container through the discharge cone, and collected them as tailings. At the same time, the heavier particles remained and settled at the bottom of the bowl as concentrate. Beneficiation was carried out continuously with a solid to liquid ratio of 30% *w/w*. The obtained gravity concentrates were combined and analyzed for gold content and elemental composition. Thereafter, the recovery percentage was calculated. Results obtained using the centrifugal separation technique were further compared with the current panning approach used at the site (10%) in order to quantify the gold recovery.

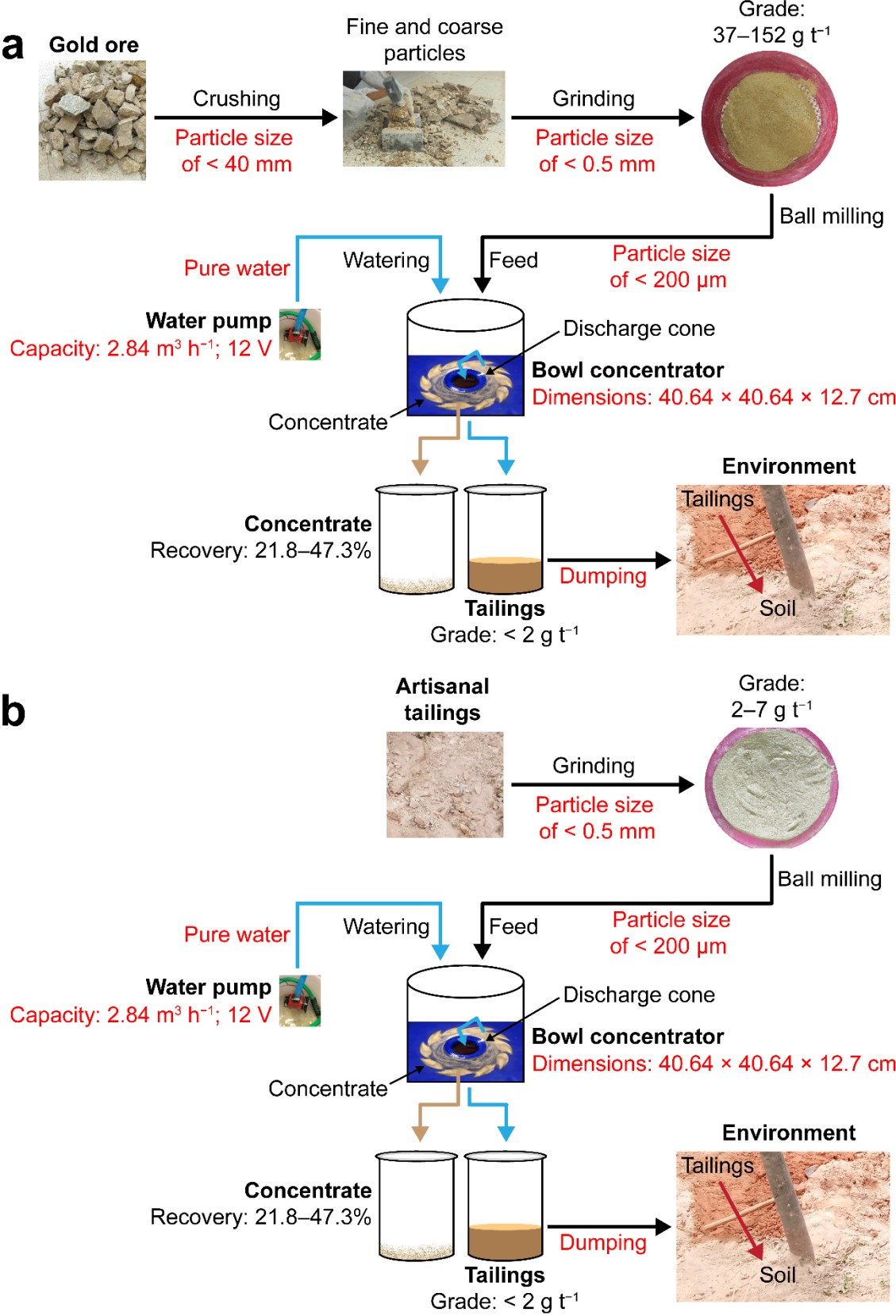

**Figure 2.** The workflow of the study indicates the importance of the centrifugal separation method in the remining of artisanal gold mine tailings and compares mineral grade and recovery percentage before and after beneficiation. (**a**) Gold ore; (**b**) artisanal tailings.

## 3. Results

### 3.1. Mineralogy of Raw Gold Ore and Resultant Artisanal Gold Mine Tailings

The gold ore particles were coarser than artisanal gold mine tailing particles (Figure 3). The diameters D10, D50, and D90 were respectively in the range of 4.8–6.8, 20.4–65.2, and 117.4–231.1 µm for gold ores, and 2.9–5.9, 18.0–56.3, and 88.7–117.2 µm for artisanal gold mine tailings. The gold grades for ores and artisanal gold mine tailings were in the range of 37–152 and 2–7g t$^{-1}$, respectively (Figure 2).

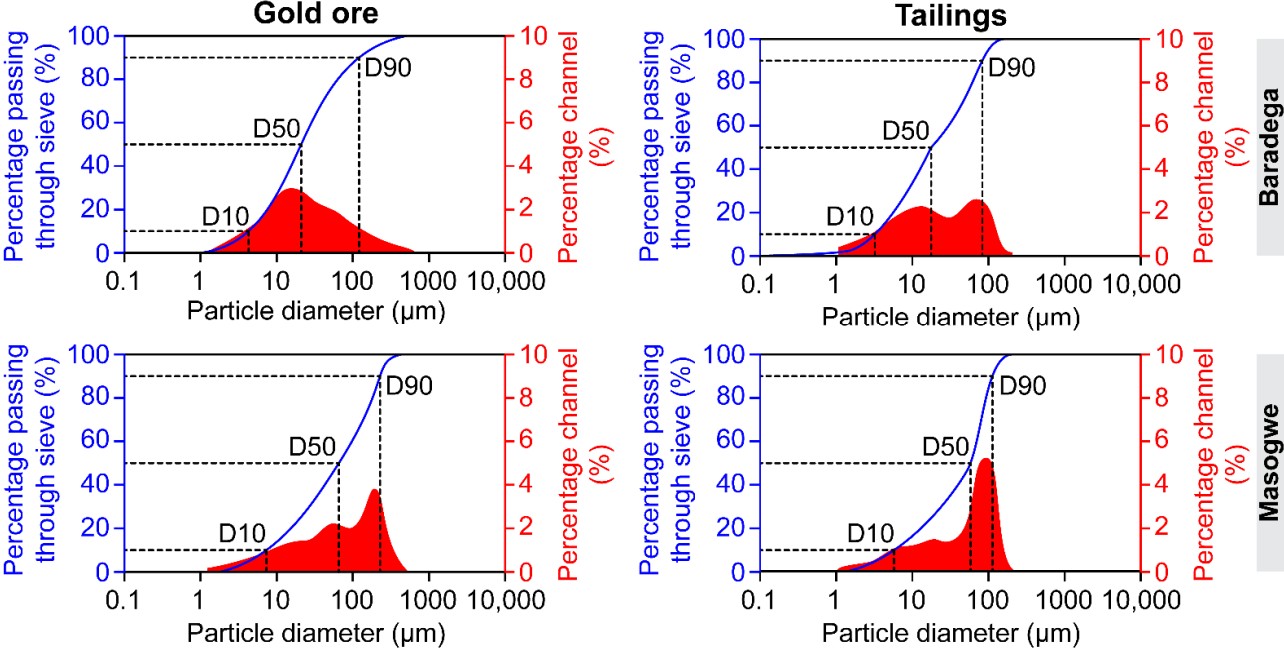

**Figure 3.** Particle size distribution for collected gold ores and resultant artisanal gold mine tailings.

Quartz (SiO$_2$) was the dominant phase in both gold ores and artisanal gold mine tailings. In addition, K-feldspar and aluminum oxide were evident in gold ores and artisanal gold mine tailings, respectively (Figure 4).

The elemental compositions revealed by the four studied samples were significantly different (Figure 5 and Table 1). Quantitative analysis of EDS results represented by the pie chart in Figure 5 showed that silicon was 34.70–44.50% in gold ores and 29.90–38.30% in tailings. Aluminium was 9.60–10.20% and 2.10–9.0%, respectively. Iron was 0.00–6.60% and 0.00–3.50%, respectively (Figure 5). Rare earth elements found in both gold ores and artisanal gold mine tailings were cerium, europium, lanthanum, neodymium, praseodymium, terbium, and yttrium. ICP-OES showed that silicon was 17.81–20.09% in gold ores and 32.42–33.81% in artisanal gold mine tailings. XRF showed that silicon was 41.06–42.71% and 38.41–39.19%, respectively (Table 1).

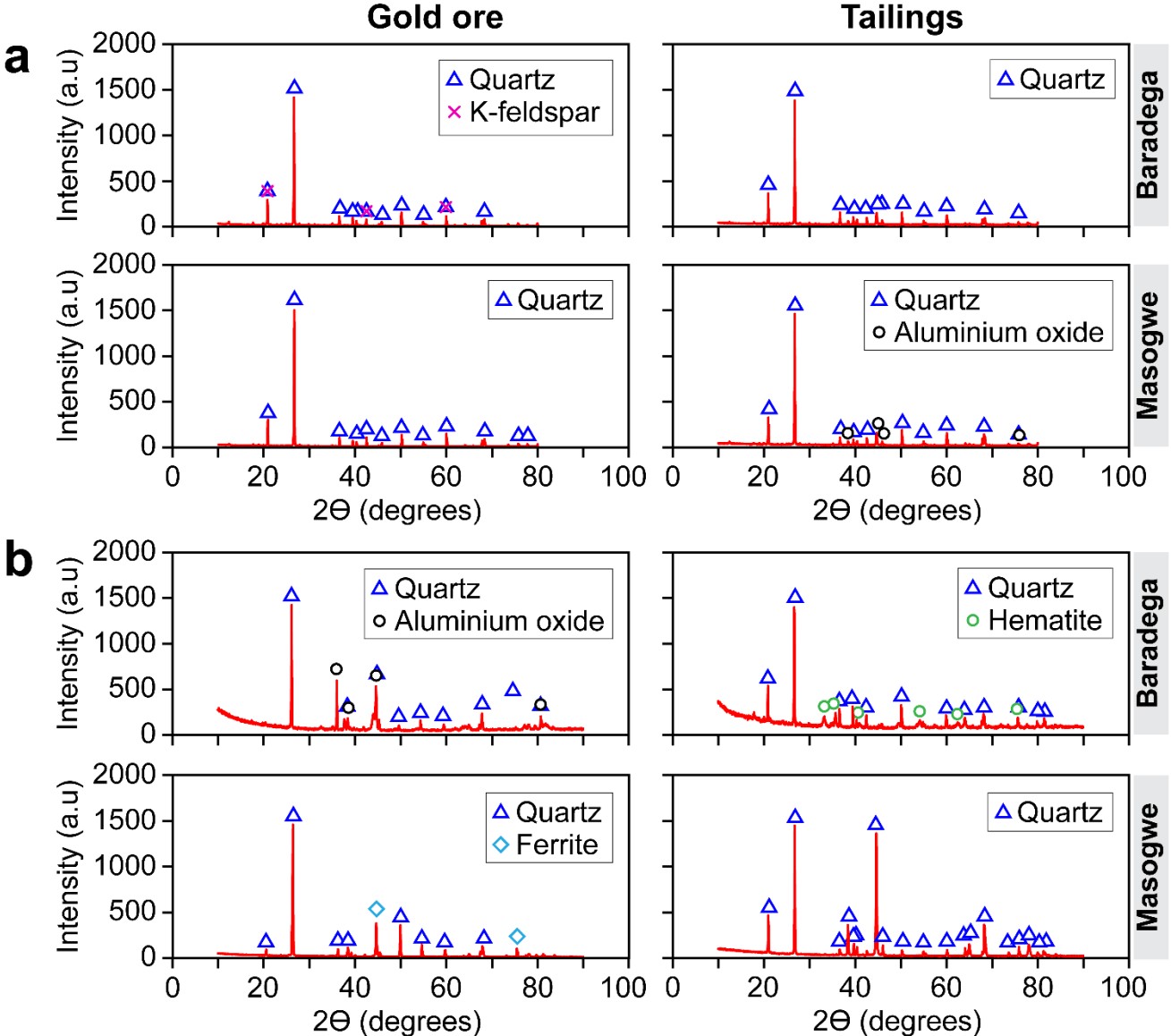

**Figure 4.** Crystallography of gold ore and resulting tailings, (**a**) before beneficiation, and (**b**) after beneficiation.

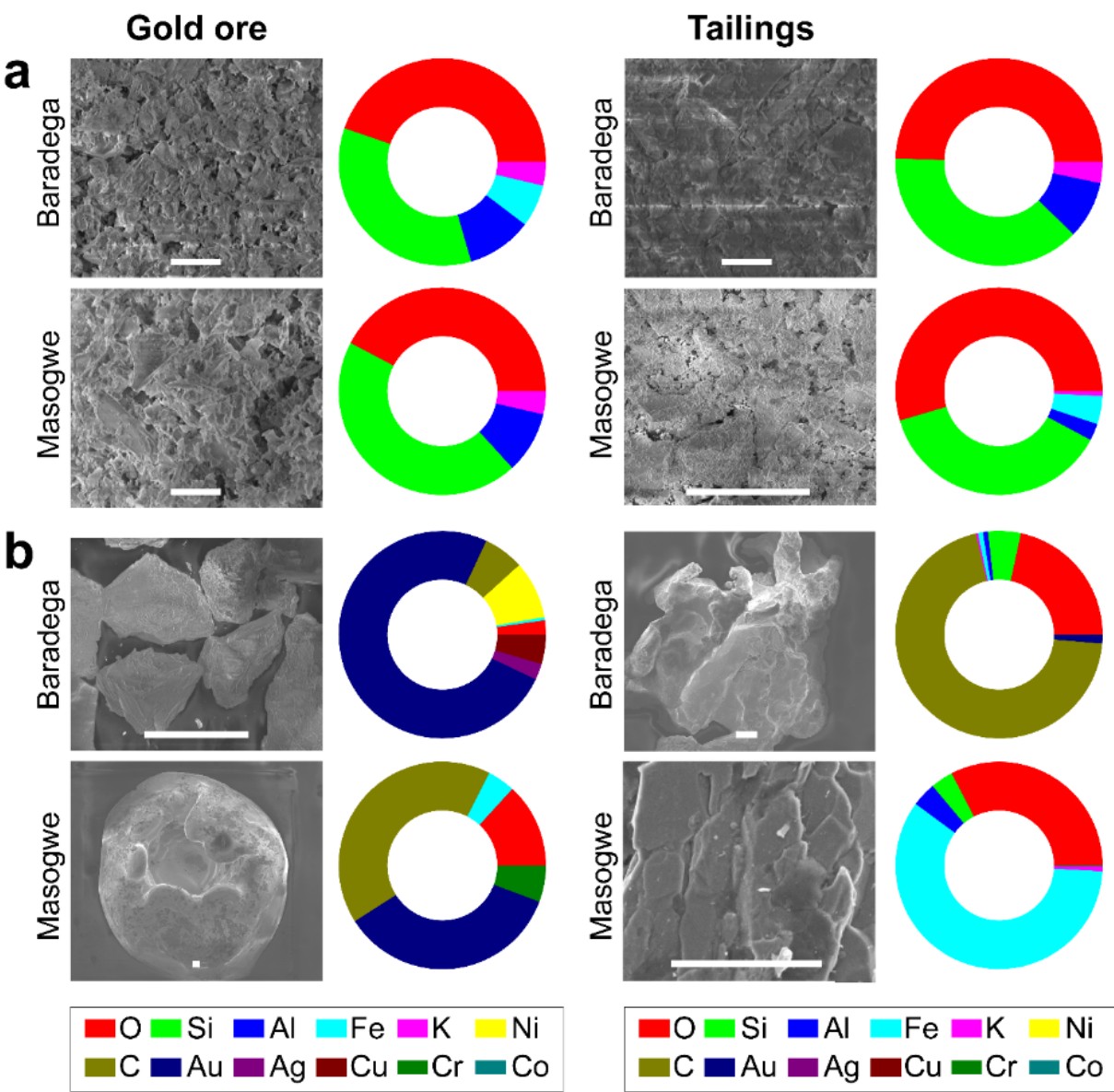

**Figure 5.** Elemental composition as measured in full scale X-ray on gold ore and resultant artisanal gold mine tailings, (**a**) before beneficiation, and (**b**) after beneficiation. The scale bars are 25 μm. The pie chart shows the proportions of 12 detect elements.

**Table 1.** Elemental composition of raw gold ore and resultant artisanal gold mine tailings.

| Element | ICP-OES | | XRF | | ICP-MS | |
|---|---|---|---|---|---|---|
| | **Gold Ore** | **Tailings** | **Gold Ore** | **Tailings** | **Gold Ore** | **Tailings** |
| **Trace element (%)** | | | | | | |
| Silicon | 17.81–20.09 | 32.42–33.81 | 41.06–42.71 | 38.41–39.19 | | |
| Potassium | 2.38–4.43 | 1.63–1.83 | 0.87–1.42 | 1.05–1.49 | | |
| Aluminium | 2.06–3.70 | 3.23–4.70 | 4.73–6.49 | 6.36–6.67 | | |
| Iron | 1.11–1.49 | 1.74–2.39 | 1.18–1.92 | 2.22–2.46 | | |
| Sodium | 0.36–0.57 | 0.44–0.46 | * | * | | |
| Magnesium | 0.10–0.11 | 0.06–0.08 | * | * | | |
| Titanium | 0.09–0.14 | 0.18–0.19 | 0.16–0.21 | 0.18–0.25 | | |
| Copper | 0.04–1.92 | 0.03–0.04 | 0.00–0.00 | 0.00–0.01 | | |
| Calcium | 0.02–0.10 | 0.02–0.03 | 0.02–0.14 | 0.00–0.02 | | |
| **Rare earth element (ppm)** | | | | | | |
| Lanthanum | | | | | 2.98–14.50 | 4.90–7.90 |
| Praseodymium | | | | | 0.60–3.00 | 1.00–1.70 |
| Yttrium | | | | | 1.00–3.30 | 0.90–1.20 |
| Europium | | | | | 0.14–0.44 | 0.19–0.40 |
| Terbium | | | | | 0.03–0.12 | 0.05–0.05 |
| Neodymium | | | | | 2.38–11.00 | 0.11–6.30 |
| Cerium | | | | | 6.08–29.00 | 9.90–15.90 |

ICP-OES, inductively coupled plasma–optical emission spectroscopy; XRF, X-ray fluorescence. "–" indicates a range; "*" shows below detection limit.

### 3.2. Mineralogy of Gold Ore and Resultant Artisanal Gold Mine Tailings Concentrate

Quartz ($SiO_2$) was the dominant phase in both gold ore and artisanal gold mine tailings concentrate. Aluminum oxide and ferrite were evident in gold ores, while hematite was observable in artisanal gold mine tailings (Figure 4). The compositions of the samples were significantly different (Figure 5 and Table 2). Based on EDS, gold was 35.30–74.90% in gold ores and 0.00–1.40% in tailings. Silicon was 0.00% and 3.50–5.10%, respectively. Aluminium was 0.00% and 0.80–3.80%, respectively. Figure 5 shows that iron was 0.50–4.30% and 0.80–59.20%, respectively. ICP-OES indicated that silicon was 44.15–41.31% for gold ores and 41.74–45.50% for artisanal gold mine tailings. XRF indicated 40.89–46.92% and 41.56–41.86%, respectively (Table 2). The gold recoveries for ores were 21.8–47.3%, while those for artisanal gold mine tailings were 46.9–63.8% (Figure 2).

**Table 2.** Elemental composition of god ore and resultant artisanal gold mine tailings concentrates.

| Element (%) | ICP-OES | | XRF | |
|---|---|---|---|---|
| | **Gold Ore** | **Tailings** | **Gold Ore** | **Tailings** |
| Silicon | 44.15–41.31 | 41.74–45.50 | 40.89–46.92 | 41.56–41.86 |
| Iron | 4.90–5.61 | 1.76–3.93 | 2.25–4.86 | 1.57–3.97 |
| Potassium | 0.46–1.13 | 0.67–1.36 | 0.06–0.61 | 0.35–0.77 |
| Copper | 0.13–0.17 | 0.14–0.18 | 0.00–0.03 | 0.00–0.00 |
| Aluminium | 0.05–0.07 | 0.21–0.28 | 2.72–2.72 | 1.82–3.19 |
| Calcium | * | * | 0.01–0.03 | 0.02–0.03 |

ICP-OES, inductively coupled plasma–optical emission spectroscopy; XRF, X-ray fluorescence. "–" indicates a range; "*" shows below detection limit.

## 4. Discussion

### 4.1. Characteristics of Artisanal Gold Mine Tailings and Potential Features as a New Source of Valued Gold

This study shows that gold associated rocks in Miyove Gold Mine allow for the extraction of gold resources efficiently (Figure 2). Manual breaking of gold ores with hammer, African stone grinding mill and old-style table shaking system are largely employed in the gold mining of Miyove (Figure 1b–d) possibly due to financial constraints on investment, lack of technology, and skilled labor [34,35]. For example, modern shaking tables that are very effective and can concentrate sizeable amounts of ore at a time, providing high-grade concentrates and liberated gold, are relatively expensive and require some skill to operate [30]. The use of primitive processing methods results in the generation of solid waste with high gold content [36]. This confirms our observation showing (1) the utilization of hammers in breaking gold ores as poor liberation of gold from the host rock [29], and as the main source of dispersed gold particles on the ground and in tailings (Figure 1b); (2) establishment of gold particle sizes in undesired ranges due to ineffectiveness in crushing and grinding gold ores (Figure 1b,c); and (3) inadequate separation of gold particles from tailings due to ineffectiveness in screening (Figure 1d).

The application of mineralogical information allows for a better understanding and solution of problems encountered during exploration and mining, and during the processing of ores, concentrates, smelter products, and related materials [13,16]. The geochemistry of artisanal gold mine tailings was conducted using various methods (XRD, EDS, ICP-MS, ICP-OES, and XRF) because gold particles and impurities in the form of rocks present in mine tailings are very tiny, and their smaller size derives amazing features, such as their chemical properties [28]. As the optimization depends on the mineralogical information of gold ore to identify the non-valuable minerals associated with gold as well as the gold host rock, quantitative analysis was performed and the information obtained provided an image of the sampled artisanal gold mine tailings behavior. Gold was hosted by quartz in the form of silicon dioxide (Figures 4 and 5 and Tables 1 and 2). The major phase identified was quartz, accompanied by minor impurities. Ferrite was identified as the phase associated with quartz, indicating the presence of iron in gold ores with small proportions of few metallic elements (Table 2). The presence of a hematite phase in the form of $Fe_2O_3$ (Figure 4) implies that gold ore and tailings samples show a paramagnetic property. Some phases detected in the studied tailings holds hazardous elements that could be considered as conveyor of human health and environment effects [37]. However, this iron-related phase was not peaked in the XRD of the gold ore sample, and possibly was masked by the other gangues presented in the ore [32,38]. The diameter of individual grains of rocks associated with gold play a major role in the processing of gold, especially when it is associated with the coarse gangue minerals [31]. Gold grades obtained after crushing and grinding all samples were high; gangues were rich in silicates (Figure 2). This suggests that processing gold ore requires specialized machinery that can be suited for the type of minerals being milled [39]. The processing of the tailings shows that about 2–7g $t^{-1}$ was lost after using traditional techniques on the sites (Figure 2), suggesting the inadequacy of methods utilized by artisanal gold miners [40]. For gold ores, the enrichment ratio (ER) ranged from 9.9 to 14.4, and for tailings, it ranged from 18.0 to 43.1.

$$ER = \frac{\text{Grade of gold in the concentrate}}{\text{Grade of gold in the feed}} \tag{1}$$

### 4.2. Importance of Centrifugal Separation in Artisanal Gold Mine Tailings Remining

Different beneficiation techniques to extract gold from ore are chosen based on various factors [41–43]. The poor processing methods posed gold losses and health risks to the miners due to the contaminants exposure. In this study, each beneficiation step was designed to increase the concentration (grade) of the valuable components of the original ore (Figure 2). Mined ore and resulting artisanal gold mine tailings underwent comminution

by crushing and/or grinding, feeding, and gravity concentration by centrifugal separation to remove the bulk of the rocks and gangue minerals. The utilization of a centrifuge was extremely beneficial as it eliminated large volumes of waste rocks from the gold. As a result, the gold in artisanal gold mine tailings was extracted at a reduced operating cost with respect to energy. The effectiveness of centrifugal separation is closely related to the gold particle size [31,39]. Additionally, particles in a centrifuge were segregated depending on their size, shape, density, and rotor speed; hence, the reported gold concentrates have enhanced grades and recoveries [15,40]. This shows the gold loss and ineffectiveness of traditional methods described in Figure 1b–d, and that mineral specific gravity facilitates pleasing separation of gold from its associated gangue minerals [28]. For example, a recovery of less than 10% of gold from solids of diameters of 74–150 μm and 40% of gold from solids of diameters of >212 μm [44].

### 4.3. Environmental and Management Implications

Artisanal and small-scale gold mining often occurs in locations where there is no large-scale mining presence [8]. Yet, artisanal small-scale mining is associated with a number of negative impacts [7,45,46]. For example, data and results indicate that poor handling of artisanal gold mine tailings can contribute to not only environmental degradation, but also abandoned pits and shafts, as well as a loss of gold resources. In addition, there is also a possibility of heavy metal contamination in the area due to poor handling processes. Previous study has examined the historical and current situation and issues of ASGM in connection to political, social, and environmental repercussions (such as population displacement, loss of livelihoods, migration of people, cost of living, water scarcity, and health implications) [47]. Thus, in this study, mineralogical characteristics of artisanal gold mine tailings using Miyove's samples in Rwanda and potential features as a new source of valued gold are shown (Figures 2–5, and Tables 1 and 2). In comparison to the current panning method used at the site (10%), the gold grades (Figure 2) reveal that the centrifugal separation technique is more effective in increasing gold recovery from artisanal gold mine tailings. This further demonstrates the need for efficiency, the use of environmentally friendly methods for gold recovery in mining (e.g., gravity methods by centrifugal separation), and artisanal gold mine tailings reuse. The priority areas for any further investigation must take into consideration more samples to sustain a successful scale-up of our findings across a wide array of mining sites and mineral separation techniques. This will thereby help to keep the artisanal and small-scale mining sector economically more productive rather than destructive.

### 5. Conclusions

This work presents mineralogical characteristics of artisanal gold mine tailings in two key mining areas of Miyove in Rwanda for potential use as a source of valuable gold using the centrifugal separation system. The results showed that artisanal gold mine tailings samples have significant amounts of gold, that can be mined using centrifugal separation practice. Quartz was a major phase, with minor impurities in both gold ores and resultant artisanal gold mine tailings. The centrifugal separation technique as applied to artisanal gold mine tailings was more beneficial in terms of gold recovery. The mineralogical characterization of artisanal gold mine tailings makes it possible to propose alternatives that improve gold ore beneficiation and the re-processing of artisanal gold mine tailings.

**Supplementary Materials:** The following supporting information can be downloaded at: https://www.mdpi.com/article/10.3390/su14138130/s1, Figure S1: Photos showing local gold processing method during samling (left) and pits excaveated by artisanal miners (right). References [48–55] are cited in the supplementary materials

**Author Contributions:** J.P.M.: investigation, formal analysis, writing—original draft, validation, writing—review and editing. J.B.H. and J.C.N.: methodology, formal analysis, writing—review and editing. L.M. and H.T.: data curation, validation, writing—review and editing. G.O.-S. and B.M.: conceptualization, methodology, resources, validation, writing—review and editing. A.R.A.: writing-review and editing. All authors have read and agreed to the published version of the manuscript.

**Funding:** This research was supported by Regional Scholarship and Innovation Fund (RSIF), a flagship program of the Partnership for Skills in Applied Sciences, Engineering and Technology (PASET), an Africa-led, World Bank-affiliated initiative.

**Institutional Review Board Statement:** Not applicable.

**Informed Consent Statement:** Not applicable.

**Data Availability Statement:** Raw data from the study is available on request.

**Acknowledgments:** The authors thank Rwanda Mines, Petroleum and Gas Board and Ngali Mining Ltd. authorizing sample collection and exportation and Miyove gold mining project for facilitating the sample collection and field investigation. The authors also thank Abdulhakeem Bello, Uwamungu Placide, and Pascaline Nyirabuhoro for their comments.

**Conflicts of Interest:** The authors declare that they have no known competing financial interests or personal relationships that could have appeared to influence the work reported in this paper.

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
