# Peer review of "Potential Uses of Artisanal Gold Mine Tailings, with an Emphasis on the Role of Centrifugal Separation Technique"

_sustainability, doi:10.3390/su14138130_

Round 1

Reviewer 1 Report

The authors developed experimental work related to sampling, characterisation and bench-scale testwork of ore and tailings samples from an artisanal mine in Rwanda. I found minor issues related to the English language and style, but have some concerns regarding the methodology, discussion and implications of the work developed. I hope you find my comments constructive.

Best regards,

The reviewer

General: I would not recommend using the hyphen in “gold-mine” throughout the text.

Line 19: What kind of effects? Please, rephrase the first sentence.

Line 34: alternative methods?

Line 36: I would suggest adding “artisanal and small-scale mining” as a keyword

Line 40: Please elaborate or review this statement and support it with appropriate references. Recent statistics show a different trend (see, for instance, https://pubs.er.usgs.gov/publication/mcs2021).

Lines 64-66: I would suggest rephrasing this sentence, e.g., “Chemical and mineralogical characterization allow a better observation of …”

Lines 66-70: Perhaps rephrase as “Grayson (2007) [28] identified about 70 different gold processing methods, except those based on cyanide or mercury, to recover gold lost by artisanal miners without the side effects of cyanide and mercury”. Also, please check the suitability of this statement. The only reference used to support it ([28]) does not seem to be a peer-reviewed journal; it can make your argument less convincing.

Lines 74-76: The sentence is unclear, please rephrase it.

Lines 83-86: I believe your research goals can be improved, e.g., perform a mineralogical characterization of artisanal gold mine tailings from a mine located in Rwanda and evaluate the technical feasibility of reprocessing this material and using it as a potential secondary source of gold.

Line 94: “supplementary information”.

Lines 99-100: The authors mention that they crushed samples manually with a hammer and then powdered them with a jaw crusher. However, the crusher does not provide “just a powder”, but probably a mix of fine and coarse particles. It is unclear how the authors dealt with the sample. My guess is that the authors screened the crusher product and then only worked with the fine fraction; if that was the case, then the authors did not work with a representative sample. In my understanding, it is expected to work with the whole sample and grind it to the required particle size distribution. The artisanal gold mining process in Figure 1 indicates that the whole material is ground before sending it to the concentration stage. Please, provide more details about your methodological procedure.

Lines 107-109: The sieve shaker uses dry sieving, but for particles below 0.075 mm, it is recommended to perform wet sieving. The authors used dry sieving up to the 0.045 mm sieve. I have concerns with the experimental procedure used by the authors.

Lines 127-129: The authors describe the procedure of centrifugal separation for the ore and tailings samples, but the type of device used in the experiments is unknown (“V_BBPP_B01”), and Figure 2 does not provide details in that regard. Please, clarify the methodological procedure and equipment employed in your research. It would be good to provide a scheme and/or photo of the device used with its inputs and outputs.   

Line 188: The authors may want to refer to silicon instead of silica

Lines 204-205: The authors state that the recoveries of gold from the primary ores were 21.8-47.3% while the recoveries from the artisanal gold mine tailings were 46.9-63.8%. Figure 5 reports the elemental composition of the samples before and after beneficiation but in a qualitative way, while the text describes some of the quantitative values. The figure clearly shows an improvement in gold grade for the ore samples after beneficiation, but this is not so evident for the tailing samples after beneficiation. Although the recoveries obtained from the tailing material were higher than those obtained from the ore, if the gold grade is too low in the concentrate obtained from the tailings sample, then it may become difficult for the artisanal miners to turn this concentrate into a saleable product or will require more concentration stages. Do the authors believe that it would be possible to recover more gold from tailings using the artisanal process, i.e. with the old-style table shaking system? Did the authors perform experimental testwork aiming to further concentrate the tailings using the current system on-site? How likely would be to scale up and improve the gold concentration system on-site based on your testwork/laboratory device? My impression is that the reported outcomes do not adequately support the conclusions.

Lines 192 and 210: It should be gold ore

Line 270: It may refer to Figures 1b-d

Line 357: It should be “Figure A1” (as cited in line 93 of the manuscript).

Pages 13-15: It is confusing to have two reference sections. Supplementary information should be best placed as an attached document in the online version of the manuscript. In this case, I believe the supplementary information should be integrated into the “Materials and Methods” section.

Author Response

We deeply appreciate your positive assessment of our manuscript and helpful suggestions. Please see the attachment with the answers and revised manuscript.

Best wishes.

Reviewer 2 Report

The article is written on a relevant topic in good technical language. It is of scientific and practical interest to specialists in the field of technological mineralogy and mining.
There are minor comments on the article.
1. The list of used sources should be expanded. There is a fairly large number of articles and materials on the processing of gold-bearing rock (both natural and technogenic).
It would be wonderful to add information (besides extraction) about product yield after centrifugal separation.
The article can be published after a minor revision.

Author Response

Many thanks for your comments and suggestions were extremely helpful for our revisions. Please see the attachment of the responses. 

With best.

Reviewer 3 Report

Dear Authors,

the subject matter described is important. Congratulations on the scientific concept and application of correct research methodology, however, the manuscript has many imperfections.

MAIN NOTES AND COMMENTS:

1.Introduction, it is worth rewriting-supplementing with specific information of the cited research

2.Fig.2, it presents rather abstract graphic - it is very simplified - not a bad approach, but not very scientific. There should be a technological diagram here

3. The concept of "particles passing through" and "chanel" in the graphs in Fig. 3 should be defined to be consistent with the international nomenclature used in mineral processing

4.specific vocabulary is used in mineral processing, e.g. benefication = enrichment, popular words should be verified for specialist ones

5.descriptions of axes in Fig. 4 should be clarified, not every potential reader performs and interprets quantitative research, it is worth understanding the content of this paper

5.All abbreviations used in the paper should be explained, either for first use or as an appendix at the end of the manuscript

6. Clear and easy to interpret graphs in Fig.5 - interesting idea, very well

7. chapter 3.3; the negative effects of artisanal exploitation have not been indicated;

8.The authors did not compare the obtained results with the results of another used gold ore enrichment technique to show that the centrifuge deposition is the most favorable

In my opinion, this manuscript has the form of a geological report and does not meet the goals set. It needs to be completed.

Kind regards, 

Reviewer

Author Response

Many thanks for your constructive comments and suggestions. Please see the attachment.

With best.

Authors

Round 2

Reviewer 1 Report

Dear authors,

I am pleased with the modifications made to the manuscript. At this second stage of revision, I believe it only requires a few more improvements. Please see my comments below:

1) Regarding my previous suggestion of removing the hyphen in "gold-mine", I recommend using the "find" tool in your text editor to check the consistency of this change throughout the text (e.g. in the title of the manuscript).

2) I spotted a typo in the legend of Figure 4 (I guess it should be "k-feldspar" instead of "k-feldspard". 

3) You have considerably improved the methodology section, and now it is clear how you handled and processed the samples. I also see more clarification in the discussion section regarding the implications of your work toward a potential upscaling of your beneficiation process on site. However, I still consider that more discussion about the sustainability of this processing approach is needed. As I understand, the custom-made Falcon Bowl concentrator receives a feed with a particle size below 200 um, more likely fed from a ball mill coupled with a classification (e.g. sieving) system. I would like that the authors discuss more on how feasible (from techno-economic, social and environmental viewpoints) would be to implement and/or upscale this new process circuit (ball mill + classification + Falcon Bowl concentrator) in the actual ASM site? Would the small-scale miners will be able to operate this circuit by themselves? Would additional training be necessary to avoid potential safety/health issues associated with the new circuit/equipment? What about the requirements for handling the remaining tailings (unrecovered material from tailings reprocessing) vs the conventional approach? From your results, it is evident that the centrifugal separation device would not operate efficiently (i.e. poor liberation and recovery of gold) without having a ball mill and size classification (e.g. sieving system). This new circuit would increase the energy and materials (e.g. grinding media) consumption and, therefore, would increase the OPEX compared to the business-as-usual case, which should be compensated by the revenue generated from reprocessing the tailings.

Author Response

Dear Reviewer thank you for the comments and suggestions. Please see the attachment for the response.

With best,

Authors

Reviewer 3 Report

Dear Authors,

thanks for answering my comments. I accept them. Glad I could help improve this paper.

I accept Fig.2 with a great deal of understanding, because the enrichment methodology is simple. However, I still think that instead of pictures I should use a professional technological scheme.

Below are some minor imperfections that require improvement:

- check the grammar in section: 2

- line 365; the equation should be given a number

Kind Regards, Reviewer

Author Response

Dear Reviewer, thank you for the comments and suggestions. Please see the attachment for the response.

With the best,

Authors
